# Assessing the knowledge, attitude and practice of osteoporosis among Pakistani women: A national social-media based survey

**Sibtain Ahmed**[1]*, **Arsala Jameel Farooqui**[1], **Nousheen Akber Pradhan**[2], **Nawazish Zehra**[1], **Hafsa Majid**[1], **Lena Jafri**[1], **Aysha Habib Khan**[1]

**1** Section of Chemical Pathology, Department of Pathology and Laboratory Medicine, Aga Khan University (AKU), Karachi City, Sindh, Pakistan, **2** Department of Community Health Sciences, Aga Khan University (AKU), Karachi City, Sindh, Pakistan

* sibtain.ahmed@aku.edu

## Abstract

### Background

There are numerous risk factors for osteoporosis and understanding and recognizing these risk factors is critical when deciding whether to take preventive measures. It is critical to reduce the healthcare expenditure burden of the Pakistani population by raising awareness and implementing osteoporosis-preventable measures. This survey aims to assess the knowledge, attitudes, and practices (KAP) of Pakistani women as well as their misconceptions about osteoporosis.

### Methods

A cross-sectional survey was conducted from August 2021 to January 2022 by the Bone & Mineral Disease research group at Section of Chemical Pathology, Department of Pathology and Laboratory Medicine, Aga Khan University, Karachi, with exemption from the ethical review committee. Using snowball sampling, a validated Osteoporosis Prevention and Awareness Tool (OPAAT) was disseminated online via social media. With informed consent, 400 Pakistani women aged ≥ 18 years were included in the study. SPSS Statistics version 25.0 was used for data analysis. Chi-square test for association and Fisher-exact test were applied, and significance level was α<0.05.

### Results

Based on the OPAAT scores of all *(n = 400)* participants, 22% *(n = 88)* had low knowledge, 44% *(n = 176)* had average knowledge, while 34% *(n = 136)* had good knowledge of osteoporosis. The most common misconceptions were about age-related risk, presentation of symptoms, radiation risk, associated risk factors like tooth loss, osteoarthritis, and knowledge about predictors of bone health.

**Data Availability Statement:** The data for this study is available at the following link: https://figshare.com/articles/dataset/Assessing_the_

Knowledge_Attitude_and_Practice_of_
Osteoporosis_among_Pakistani_Women_a_
National_Social-Media_Based_Survey/22657006

**Funding:** The authors received no funding for this work.

**Competing interests:** The authors have declared that no competing interests exist.

## Conclusion

Adult Pakistani women have a fair understanding of osteoporosis, but the OPAAT tool clarifies some common misconceptions. There is a need to develop educational strategies to increase the knowledge of osteoporosis among Pakistani adults and to promote a bone-healthy lifestyle.

## Introduction

Pakistan, being the world's sixth most populated state, currently has more than 8 million older adults and this number is expected to reach 27 million by 2050 [1]. Osteoporosis is more likely to occur in adults aged over 50, leading to a high fracture risk [2], and fragility fractures are associated with significant morbidity and mortality in the elderly. Being a silent disease, symptoms do not appear until the disease has advanced, and for many people, the condition usually goes undetected until a fracture occurs [2, 3].

Although osteoporosis leads to deterioration in quality of life and quality-adjusted life-year, it can be adequately improved with treatment [4]. Despite this, data shows that in general it stays underdiagnosed and undertreated in Pakistan [3]. Reasons for under diagnosis include lack of awareness among the public and professionals, the asymptomatic nature of osteoporosis, inaccessibility to care, inadequate dual-energy X-ray absorptiometry (DEXA) machines, and high cost of diagnostics [4]. There are numerous risk factors for osteoporosis and osteoporosis-related fractures and understanding and recognizing these risk factors is critical for preventive measures.

In Pakistan, 0.4% of the total budget is allocated to health, and osteoporosis, as a preventable condition, imposes a significant financial burden. People in Pakistan face numerous barriers to accessing healthcare due to inadequately allocated resources. However, access to healthcare begins with the individual being aware of what they require and what their rights as a common citizen are. Knowledge of osteoporosis plays an important role in developing attitudes towards the disease prevention which in turn impacts health care behaviors [5]. Several studies in different populations have assessed the knowledge and attitudes toward osteoporosis aiming at providing baseline data essential for planning educational interventions in this topic [6], but data in Pakistan is scattered. It is critical to reduce the Pakistani population's healthcare expenditure burden by increasing awareness and implementing osteoporosis-preventable measures [3, 7].

The aim of this survey is to evaluate the knowledge, attitudes, and practices (KAP) of osteoporosis among Pakistani women aged ≥ 18 years and to evaluate their misconceptions regarding osteoporosis.

## Methodology

### Study design and study setting

This cross-sectional survey was conducted by the Bone & Mineral Disease Research Group at Section of Chemical Pathology, Department of Pathology and Laboratory Medicine, the Aga Khan University, Karachi after exemption from ethical review committee (2021-6459-18411) over a period of six months from August 2021 to January 2022.

## Sample size

An open EPI calculator at 95% confidence interval was used which yielded a sample of 384. This sample size was calculated on the assumption that 50% of participants possess some knowledge of osteoporosis [4]. However, for statistical convenience we recruited 400 subjects.

## Sampling technique and participants

Participants were recruited via snowball technique, if they met the following inclusion criteria: female gender, Pakistani national residing in Pakistan, aged $\geq$ 18 years, ability to read English, having the ability to provide written informed consent and having access to smart devices with basic IT skills to complete the survey. Unlike previous Pakistani KAP studies, this research was not targeted towards medical professionals [8].

## Primary and secondary outcome variables

Primary outcome variable: To evaluate the knowledge, attitudes, and practices (KAP) of osteoporosis among Pakistani women aged $\geq$ 18 years

Secondary outcome variable: To evaluate their misconceptions regarding osteoporosis.

## Exposure of interest

A validated Osteoporosis Prevention and Awareness Tool (OPAAT) [9] was used to assess participants' knowledge and awareness about osteoporosis. This OPAAT is available in English and has 30 questions assessing knowledge about osteoporosis pathophysiology, prevention, and consequences of untreated disease. Surveys were graded as 1 point for each correct answer and 0 point for incorrect answers or if the participant chose the 'don't know' option. A score of $\geq$ 24 out of 30, 19–23 and < 19 on the OPAAT was considered good, average, and low knowledge respectively.

An OPAAT containing e-questionnaire was developed using Google Docs and administered online via various social media groups (i.e. Facebook, Twitter, Telegram, LinkedIn), email groups and WhatsApp groups that included women $\geq$ 18 years by the study investigators.

## Ethics

Electronic written consent for participation was taken at the initial page of the survey. The participation in the survey was completely voluntary and any person could opt out and withdraw by not submitting the answers. To ensure confidentiality any personal details were anonymized, and study identifiers were allotted. Additionally, all data was stored in a password-protected electronic format.

## Data management

We obtained informed consent, assured anonymity, and clearly communicated our data handling practices protecting participants' privacy. Robust security measures, such as encrypted connections and secure storage systems, were implemented to prevent unauthorized access or breaches of the data. The data was carefully stored, and a retention period of 7 years was determined according to Section 4.3 of AKU's Policy on Code of Good Research Practice and Access to Participants Data [10]. Validation checks and data cleaning techniques were utilized to maintain data quality and integrity. Respect for participant control was upheld, offering options for access, modification, or deletion of their data upon request.

## Statistical analysis

Information from the questionnaires was de-identified and transcribed into Microsoft Excel for ease of data analysis and assessed for completion. Data was analysed using Chi-square analysis in IBM SPSS Statistics version 25.0 (IBM Corp, Armonk, NY, USA). A p-value < 0.05 was considered statistically significant.

## Results

All 400 respondents completed the survey. Based on the OPAAT scores, 22% (n = 88) had low knowledge of osteoporosis, 44% (n = 176) had average knowledge, while 34% (n = 136) had good knowledge of osteoporosis. Higher knowledge scores were significantly associated with education level (p = 0.017), and employment status (0.002), and province of residence (<0.0001), with Sindh yielding the maximum number of responses. Fig 1 shows the country-wide distribution of study participants. Table 1 shows the demographic characteristics of the study participants. Table 2 shows the participants' scores on the Osteoporosis Prevention and Awareness Tool with correct answers. Table 3 depicts the evaluation of the association between different variables and participants knowledge.

## Discussion

The Health Belief Model, implemented in a study by Al-Muraikhi et al, asserts that to plan a successful educational intervention, there should be knowledge about the target group's perceptions regarding susceptibility, the severity of the condition, benefits of taking certain actions to reduce the risk, barriers (e.g., costs of the advised action), and cues to action (strategies for activating the "readiness" to undertake health actions) [11]. To date, little is known about the perceptions, awareness, and knowledge of osteoporosis among the Pakistani

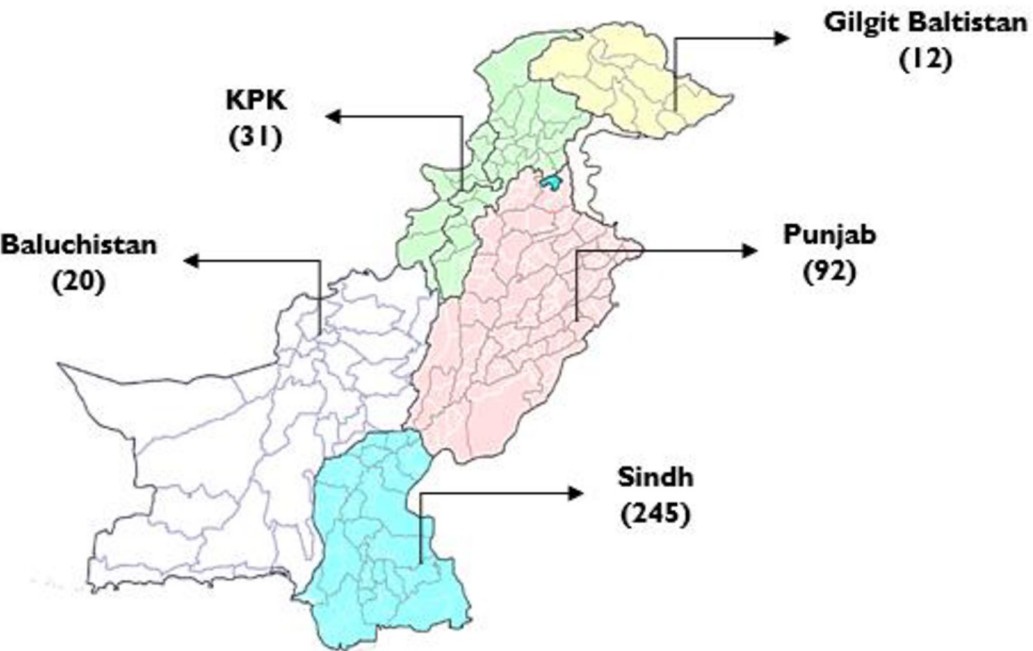

**Fig 1. Province-wise distribution of study participants.**

**Table 1. Baseline data of OPAAT survey respondents (n = 400).**

| Characteristics | | Frequency | % |
|---|---|---|---|
| Age (years) | 18–30 | 115 | 28.7% |
| | 31–40 | 152 | 38.0% |
| | 41–50 | 59 | 14.8% |
| | 51–60 | 49 | 12.3% |
| | >60 | 24 | 6.3% |
| Province | Sindh | 245 | 61.3% |
| | Punjab | 92 | 23.0% |
| | Gilgit Baltistan | 12 | 3.0% |
| | KPK | 31 | 7.8% |
| | Baluchistan | 20 | 5.0% |
| Marital Status | Single | 118 | 29.5% |
| | Married | 256 | 64.0% |
| | Separated / Divorced | 14 | 3.6% |
| | Widowed | 12 | 3.0% |
| Education Level | Secondary | 33 | 8.3% |
| | Bachelors | 181 | 45.3% |
| | Masters | 169 | 42.3% |
| | PhD | 17 | 4.3% |
| Employment Status | Employed | 167 | 41.8% |
| | Self-employed | 52 | 13.0% |
| | Looking for a job | 59 | 14.8% |
| | Housewife | 106 | 26.5% |
| | Retired | 16 | 4.0% |
| Post Menopausal Status | Yes | 111 | 27.8% |
| | No | 289 | 72.3% |
| OPAAT score | Low knowledge | 88 | 22.0% |
| | Average knowledge | 176 | 44.0% |
| | Good knowledge | 136 | 34.0% |

OPAAT: Osteoporosis Prevention and Awareness Tool

population, especially women, with only a handful of studies highlighting the misconceptions in the public [7, 12, 13].

In our study, higher knowledge scores were significantly associated with education level, employment status, and province of residence. This indicates that individuals with higher education levels and those who were employed tended to have better knowledge of osteoporosis. These findings are like a study by Din et al, that recommended that the improvement of socio-economic status and increasing the level of education could play a vital role in reducing the risks of osteoporosis [14]. Additionally, the province of Sindh yielded the maximum number of responses, suggesting a potential regional variation in knowledge levels, the reasons for which are yet to be explored.

Many of the participants (27.8%) reported being post-menopausal, which is a crucial factor in understanding the relevance of osteoporosis and its management. A 2019 study by Senthilraja et al demonstrated a substantial lack of knowledge and awareness regarding osteoporosis among postmenopausal women in India [15]. However, a 2021 study by Abdolalipour et al concluded that Education and the adoption of health-promoting lifestyle practices are crucial in preventing primary osteoporosis and enhancing the quality of life for postmenopausal women [16].

**Table 2. Scores of osteoporosis prevention and awareness tool with correct answer (n = 400).**

| Question | Correct | | Incorrect | |
|---|---|---|---|---|
| | f | % | f | % |
| 1. Osteoporosis makes bone weaker, more brittle, and more likely to break (fracture). (True) | 396 | 99% | 4 | 1% |
| 2. Everybody will get osteoporosis as it is part of ageing. (False) | 174 | 44% | 226 | 57% |
| 3. Osteoporosis occurs because bone is removed faster than it is formed. (True) | 277 | 69% | 123 | 31% |
| 4. Osteoporosis and osteoarthritis are different names we use to describe the same disease. (False) | 260 | 65% | 140 | 35% |
| 5. Osteoporosis usually has no symptoms. (True) | 163 | 41% | 237 | 59% |
| 6. Postmenopausal women are not at risk for osteoporosis. (False) | 344 | 86% | 56 | 14% |
| 7. Osteoporosis is an untreatable disease. (False) | 267 | 67% | 133 | 33% |
| 8. A bone mineral density test is used to diagnose osteoporosis. (True) | 358 | 90% | 42 | 11% |
| 9. I do not need a bone mineral density test unless I fracture my bones. (False) | 308 | 77% | 92 | 23% |
| 10. A bone mineral density test is high in radiation. (False) | 197 | 49% | 203 | 51% |
| 11. A bone mineral density test should be performed monthly to monitor bone loss. (False) | 253 | 63% | 147 | 37% |
| 12. Osteoporosis results in back pain. (True) | 336 | 84% | 64 | 16% |
| 13. Osteoporosis results in loss of height or hunchback. (True) | 248 | 62% | 152 | 38% |
| 14. Osteoporosis results in loss of mobility (unable to move around by myself). (True) | 315 | 79% | 85 | 21% |
| 15. Osteoporosis results in tooth loss. (False) | 213 | 53% | 187 | 47% |
| 16. Osteoporosis results in joint pain or swelling of fingers. (False) | 128 | 32% | 272 | 68% |
| 17. The recommended daily intake for calcium in women is 1,000 mg. (True) | 345 | 86% | 55 | 14% |
| 18. It is too late to increase calcium intake after age 18 years. (False) | 316 | 79% | 84 | 21% |
| 19. Glucosamine can help prevent osteoporosis. (False) | 217 | 54% | 183 | 46% |
| 20. Calcium supplements can help prevent osteoporosis. (True) | 374 | 94% | 26 | 7% |
| 21. The regular dose of calcium supplements can cause kidney stones. (False) | 222 | 56% | 178 | 45% |
| 22. Food such as milk, yogurt, cheese, yellow dhal and spinach are rich in calcium. (True) | 386 | 97% | 14 | 4% |
| 23. You can obtain your recommended daily intake of vitamin D via exposing your skin to sunlight for about 15 minutes a day. (True) | 343 | 86% | 57 | 14% |
| 24. Increasing coffee and tea intake can help in osteoporosis prevention. (False) | 301 | 75% | 99 | 25% |
| 25. Weight-bearing exercise (such as brisk walking and line dancing) can decrease bone loss. (True) | 271 | 68% | 129 | 32% |
| 26. Exercise will wear out bones. (False) | 291 | 73% | 109 | 27% |
| 27. Certain medications (such as sleeping tablets or high blood pressure medications) may reduce the risk of falling. (False) | 275 | 69% | 125 | 31% |
| 28. To prevent falls, comfortable shoes with a good grip should be used. (True) | 377 | 94% | 23 | 6% |
| 29. Poor vision may lead to falls. (True) | 363 | 91% | 37 | 9% |
| 30. Being underweight helps prevent osteoporosis. (False) | 262 | 66% | 138 | 35% |

With these results showing receptiveness in the public, we are presented with a promising area to plan interventions, which we can tackle by understanding their misconceptions.

## Misconceptions about osteoporosis

The misconceptions present in more than half of the study participants are highlighted below. Many of the participants believed that osteoporosis is a natural consequence of aging. Although this claim has been refuted by many researchers [17, 18], the public opinion about it stays unchanged. According to some researchers, elderly people who have osteoporosis-related fractures should not be considered "average elderly," but rather frail, signifying decreased musculoskeletal functioning [19–21]. Though medical literature is clear about the difference between the two, the public perspective remains fuzzy. A 2018 review by Chan et al showed

**Table 3. Association of baseline data with participants' knowledge of osteoporosis prevention and awareness.**

| Variables | Low knowledge (a) | Average knowledge (b) | High knowledge (c) | p-value |
|---|---|---|---|---|
| | n (%) | n (%) | n (%) | |
| **Age (years)** | | | | 0.297 |
| 18–30 | 26(29.5) | 47(26.7) | 42(30.9) | |
| 31–40 | 41(46.6) | 63(35.8) | 48(35.3) | |
| 41–50 | 12(13.6) | 30(17.0) | 17(12.5) | |
| 51–60 | 8(9.1) | 23(13.1) | 18(13.2) | |
| >60 | 1(1.1) | 13(7.4) | 11(8.1) | |
| **Province** | | | | <0.0001* |
| Sindh | 29(33.0) | 115(65.3) [a] | 101(74.3) [a] | |
| Punjab | 27(30.7) | 37(21.0) | 28(20.6) | |
| KPK | 21(23.9) [b, c] | 6(3.4) | 4(2.9) | |
| Baluchistan | 9(10.2) [c] | 9(5.1) | 2(1.5) | |
| Gilgit Baltistan | 2(2.3) | 9(5.1) | 1(0.7) | |
| **Marital Status[†]** | | | | >0.999 |
| Single | 33(37.5) | 46(26.1) | 39(28.7) | |
| Married | 48(54.5) | 116(65.9) | 92(67.6) | |
| Separated / Divorced | 3(3.4) | 11(6.3) | 0(0.0) | |
| Widowed | 4(4.5) | 3(1.7) | 5(3.7) | |
| **Education Level** | | | | 0.017* |
| Secondary | 14(15.9) [c] | 13(7.4) | 6(4.4) | |
| Bachelors | 45(51.1) | 78(44.3) | 58(42.6) | |
| Masters | 26(29.5) | 79(44.9) [a] | 64(47.1) [a] | |
| PhD | 3(3.4) | 6(3.4) | 8(5.9) | |
| **Employment Status** | | | | 0.002* |
| Employed | 26(29.5) | 73(41.5) | 68(50.0) [a] | |
| Self-employed | 11(12.5) | 20(11.4) | 21(15.4) | |
| Looking for a job | 14(15.9) | 24(13.6) | 21(15.4) | |
| Housewife | 36(40.9) [c] | 51(29.0) [c] | 19(14.0) | |
| Retired | 1(1.1) | 8(4.5) | 7(5.1) | |
| **Post Menopausal Status** | | | | 0.629 |
| Yes | 27(30.7) | 50(28.4) | 34(25.0) | |
| No | 61(69.3) | 126(71.6) | 102(75.0) | |

* Chi-square test for association was applied and Significance level was α<0.05.

[†] Fisher-exact test was applied, and Significance level was α<0.05.

[a, b, c] Tests are adjusted for all pairwise comparisons within a row of each innermost subtable using the Bonferroni correction. Significance level for post-hoc test was α<0.05.

similar misconceptions about the condition in a population of adolescents and young adults, suggesting that awareness strategies should be focused on this group [22].

Most of our population assumed that osteoporosis presented with visible symptoms, a dangerous belief that must be dispelled. The condition manifests silently, staying undetected until fractures occur, or there is back pain caused by a fractured or collapsed vertebra, loss of height, or stooped posture [19]. Low bone mass is the most important risk factor for osteoporosis, and it alone predicts fracture risk. Furthermore, biologic risk factors (such as increasing age, a family history of fracture, and race) as well as lifestyle risk factors (such as low calcium intake and vitamin D deficiency) may increase the risk of osteoporosis and osteoporosis-related fractures [21].

An interventional study from Sobeih and El-Wahed in 2018 studied the knowledge of 60 Egyptian women before and after a three-part educational intervention (interviewing questionnaire sheet, perception questionnaire sheet, and women's complaint tool), reporting a statistically significant change in the participants' attitudes and beliefs [23].

Many of the participants also had misconceptions about the diagnostic tests involved, falsely assuming that a bone mineral density (BMD) test is high in radiation. BMD is usually detected via dual energy X-ray absorptiometry (DEXA), high-resolution peripheral quantitative CT (HR-pQCT), peripheral quantitative CT (pQCT), quantitative CT (QCT), or ultrasonography (QUS). All the above technologies except for QCT expose the user to minimal radiation risk [24, 25]. A false perception about this may promote hesitation in getting tested, and specific informational guidelines can help dispel this fear.

Nearly half of the respondents also believed that osteoporosis results in tooth loss. While the two are related in terms of bone loss, periodontitis is an inflammatory condition of the periodontium (the surrounding structure of the tooth) that results in bone loss due to clinical attachment loss. Its pathogenesis is not related to bone fragility as seen in osteoporosis and may occur independently of the latter [26]. Opinions about the associations of the two conditions remain contradictory, but there is consensus on low BMD being a risk factor for periodontitis. Shared risk factors for both diseases include age, genetics, hormonal change, smoking, calcium, and vitamin D deficiency, as well as each other [26, 27].

Osteoporotic bone is weak and fragile but does not lead to joint pain or swelling of fingers. These symptoms are characteristic of osteoarthritis, which is an inflammation of the joints, usually developing with age and resulting in bone degeneration. An overwhelming majority of our respondents falsely assumed that osteoporosis presents with these symptoms [28].

Participants also held some misconceptions regarding the predictors of bone health. Most of the participants believed that glucosamine could help prevent osteoporosis. This is a claim that needs more understanding [29], and there is a need to enhance knowledge about the role of glucosamine in bone health. They also believed that regular doses of calcium supplements can cause kidney stones. This depends upon the quantity of calcium consumed via diet and supplementation. Recommended daily allowance varies according to age and risk, and a study even found that high calcium levels in the serum are inversely relative to the risk of osteoarthritis in the knee [30].

## Strengths

The fact that this was the first study in Pakistan to undertake a social media poll on the subject and one of the first to capture the perceptions of the public rather than just healthcare professionals are the study's strengths.

## Limitations

Although this study's results are encouraging, its sampling strategy has certain drawbacks. Our cohort might not be typical of the broader population because the data was collected using social media and we did not target women who do not have access to social media, such as the impoverished, physically challenged, or those with low levels of education. The survey also reveals a high percentage of scores in urban Sindh, where the presence of more advanced technology may affect the percentage results. This study should serve as a starting point for future, more in-depth research on Pakistani demographics.

## Conclusion

Pakistani women's understanding of osteoporosis is moderate, but the OPAAT tool helps identify misconceptions. Understanding the demographic characteristics and their association with knowledge can help in designing tailored educational campaigns and interventions to promote better knowledge, attitudes, and practices related to osteoporosis in the Pakistani public women Identifying at-risk individuals is crucial for reducing morbidity and mortality from osteoporotic fractures. Addressing misconceptions requires individual-level interventions like school education, exercise, and sunlight exposure for long-term bone health. Social media is a valuable tool for information gathering and targeting specific populations. Further research is necessary to explore the long-term effects of these interventions on Pakistani communities.

## Author Contributions

**Conceptualization:** Sibtain Ahmed, Hafsa Majid, Aysha Habib Khan.

**Data curation:** Sibtain Ahmed, Hafsa Majid.

**Formal analysis:** Sibtain Ahmed, Nousheen Akber Pradhan, Aysha Habib Khan.

**Investigation:** Sibtain Ahmed, Nousheen Akber Pradhan.

**Methodology:** Sibtain Ahmed, Nousheen Akber Pradhan, Hafsa Majid.

**Project administration:** Sibtain Ahmed, Hafsa Majid, Aysha Habib Khan.

**Resources:** Aysha Habib Khan.

**Software:** Sibtain Ahmed.

**Supervision:** Sibtain Ahmed, Lena Jafri, Aysha Habib Khan.

**Validation:** Sibtain Ahmed, Nousheen Akber Pradhan, Nawazish Zehra, Lena Jafri.

**Visualization:** Sibtain Ahmed, Arsala Jameel Farooqui, Nawazish Zehra, Hafsa Majid, Lena Jafri, Aysha Habib Khan.

**Writing – original draft:** Sibtain Ahmed, Arsala Jameel Farooqui, Aysha Habib Khan.

**Writing – review & editing:** Sibtain Ahmed, Arsala Jameel Farooqui, Nawazish Zehra, Hafsa Majid, Lena Jafri, Aysha Habib Khan.

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
