## [Decision Letter · Decision Letter 0]

13 Mar 2023

PONE-D-22-34888Assessing the Knowledge, Attitude and Practice of Osteoporosis among Pakistani Women: A National Social-Media Based SurveyPLOS ONE

Dear Dr. Ahmed,

Thank you for submitting your manuscript to PLOS ONE. After careful consideration, we feel that it has merit but does not fully meet PLOS ONE’s publication criteria as it currently stands. Therefore, we invite you to submit a revised version of the manuscript that addresses the points raised during the review process.

We look forward to receiving your revised manuscript.

Kind regards,

Tauqeer Hussain Mallhi, Ph.D

Academic Editor

PLOS ONE

Journal Requirements:

http://www.smj.org.sg/sites/default/files/SMJ-62-190.pdf and

https://cyberleninka.org/article/n/721573

In your revision ensure you cite all your sources (including your own works), and quote or rephrase any duplicated text outside the methods section. Further consideration is dependent on these concerns being addressed

3. Please provide additional details regarding participant consent. In the ethics statement in the Methods and online submission information, please ensure that you have specified what type you obtained (for instance, written or verbal, and if verbal, how it was documented and witnessed). If your study included minors, state whether you obtained consent from parents or guardians. If the need for consent was waived by the ethics committee, please include this information

Reviewers' comments:

Reviewer's Responses to Questions

**Comments to the Author**

1. Is the manuscript technically sound, and do the data support the conclusions?

Reviewer #1: Yes

Reviewer #2: Yes

Reviewer #3: Yes

Reviewer #4: Yes

2. Has the statistical analysis been performed appropriately and rigorously? 

Reviewer #1: Yes

Reviewer #2: Yes

Reviewer #3: Yes

Reviewer #4: Yes

3. Have the authors made all data underlying the findings in their manuscript fully available?

Reviewer #1: Yes

Reviewer #2: Yes

Reviewer #3: Yes

Reviewer #4: Yes

4. Is the manuscript presented in an intelligible fashion and written in standard English?

Reviewer #1: Yes

Reviewer #2: Yes

Reviewer #3: Yes

Reviewer #4: Yes

5. Review Comments to the Author

Reviewer #1: 1. It would be better to define primary and secondary outcome variables in methods section in line with objectives. Results, discussion and conclusion need to follow the same in chain.

2. Methods sections is supposed to be core of any study. Here, methods section contains inadequate information. For example, following components for methods section need to be well described.

i. Details of Study design

ii. Setting

iii. Sample size estimation

iv. Sampling technique

v. Participant

vi. Primary and secondary outcome variables with working definition

vii. Intervention/issue of interest (exposure)

viii. Ethics and end point

ix. Statistical analysis

3. The discussion section needs to be described scientifically. Kindly frame it along the following lines:

a. Main findings of the present study

b. Comparison with other studies

c. Implication and explanation of findings

d. Strengths and limitations

e. Conclusion, recommendation and future directions

Reviewer #2: This research found that adult Pakistani women have a fair understanding of osteoporosis, but the OPAAT tool clarifies some common misconceptions. There is a need to develop educational strategies to increase the knowledge of osteoporosis among Pakistani adults and to promote a bone-healthy lifestyle.This is an interesting and meaningful study, and I recommend accept.

Reviewer #3: Thank you for the article.

It was a nice read. Obviously, the effort was meant to highlight the issue and find out the misconceptions using validated outcome measure. The authors have just done that. In a survey, there will always be issues with demographics and how was the population selected. Urban Sindh being more tech advanced and educated obviously will sway the percentages in any survey. If the scores could be adjusted according to provinces, this would have been cherry on top.

Best Regards

Reviewer #4: Assessing the Knowledge, Attitude and Practice of Osteoporosis among Pakistani

Women: A National Social-Media Based Survey

Review

Thank you for letting me review your article. This study assesses Pakistani women's knowledge, attitudes, practices about osteoporosis by using OPAAT. The content is useful for educating Pakistanis about osteoporosis in the future.

Methods

Line70-71 It is not clear on how you arrived at the sample size, please describe what you estimated the absolute precision and expected response rate to be.

Data management is extremely important for research using social media, as in this study. Please describe in detail how you maintained the quality of the data.

Results

Line 78 What was the response rate? Does it mean it was 100%? Please state.

Discussion

Please provide more details on strength and limitation, such as how limitation may not apply to the general public. In terms of limitation, since the study participants were only social media users, the findings cannot be generalized to the general population.

Please describe what proposed solution is possible based on the results of this study. It would be a very useful paper if specific solutions could be mentioned that would raise the level of knowledge of osteoporosis among the people of Pakistan.

6. PLOS authors have the option to publish the peer review history of their article (what does this mean?). If published, this will include your full peer review and any attached files.

Reviewer #1: **Yes: **Dr. Satish Prasad Barnawal

Reviewer #2: No

Reviewer #3: No

Reviewer #4: No

<quillbot-extension-portal></quillbot-extension-portal>

---

## [Author Response · Author response to Decision Letter 0]

19 Apr 2023

Response to Reviewers:

Review Comments to the Author

Reviewer #1: 1. It would be better to define primary and secondary outcome variables in methods section in line with objectives. Results, discussion and conclusion need to follow the same in chain.

Response: Primary and secondary outcomes are now defined in the Methods. The rest of the sections also follow the same in chain.

2. Methods sections is supposed to be core of any study. Here, methods section contains inadequate information. For example, following components for methods section need to be well described.

i. Details of Study design

ii. Setting

iii. Sample size estimation

iv. Sampling technique

v. Participant

vi. Primary and secondary outcome variables with working definition

vii. Intervention/issue of interest (exposure)

viii. Ethics and end point

ix. Statistical analysis

Response: The methods are now well defined and in line with the format mentioned above. 

3. The discussion section needs to be described scientifically. Kindly frame it along the following lines:

a. Main findings of the present study

b. Comparison with other studies

c. Implication and explanation of findings

d. Strengths and limitations

e. Conclusion, recommendation and future directions

Response: Thank you for your feedback. We have modified the section according to your guidance and made changed in the section in the revised manuscript.

Reviewer #2: This research found that adult Pakistani women have a fair understanding of osteoporosis, but the OPAAT tool clarifies some common misconceptions. There is a need to develop educational strategies to increase the knowledge of osteoporosis among Pakistani adults and to promote a bone-healthy lifestyle. This is an interesting and meaningful study, and I recommend accept.

Reviewer #3: Thank you for the article.

It was a nice read. Obviously, the effort was meant to highlight the issue and find out the misconceptions using validated outcome measure. The authors have just done that. In a survey, there will always be issues with demographics and how was the population selected. Urban Sindh being more tech advanced and educated obviously will sway the percentages in any survey. If the scores could be adjusted according to provinces, this would have been cherry on top.

Best Regards

Response: Thank you for such a wonderful review, we have tried our best to highlight these disparities in our text, but it was not a focus of our objective hence it is too late to adjust our results now. We have mentioned it in the Limitations section. 

Reviewer #4: Assessing the Knowledge, Attitude and Practice of Osteoporosis among Pakistani

Women: A National Social-Media Based Survey

Review

Thank you for letting me review your article. This study assesses Pakistani women's knowledge, attitudes, practices about osteoporosis by using OPAAT. The content is useful for educating Pakistanis about osteoporosis in the future.

Methods

Line70-71 It is not clear on how you arrived at the sample size, please describe what you estimated the absolute precision and expected response rate to be.

Response: 

For sample size, an Open EPI calculator at 95% confidence interval was used which yielded a sample of 384. This sample size is calculated on the assumption that 50% of participants possess some knowledge of osteoporosis (6). However, for statistical convenience we recruited 400 subjects.

6. Lulla D, Teo CW, Shen X, Loi ZB, Quek KW, Lis HL, Koh SA, Chan ET, Lim SW, Low LL. Assessing the knowledge, attitude and practice of osteoporosis among Singaporean women aged 65 years and above at two SingHealth polyclinics. Singapore Medical Journal. 2021 Apr 1;62(4):190-4.

Data management is extremely important for research using social media, as in this study. Please describe in detail how you maintained the quality of the data.

Response: 

To ensure the quality of the data, we ensured that the following steps were undertaken.

• Research question and survey question was well-defined prior to the collection of the data.

• Easily comprehensible OPAAT tool was used. The other surveys in which this tool was used reported good understanding of the questions from the participants’ end.

• The forms were disseminated ethically and transparently through social media where our target population was likely to be found, ensuring a variety of respondents.

• The data was fully checked and assessed for inconsistencies, errors and incompletion. If found, this missing data was supposed to be removed from the data set, which did not happen in our case as all the responses were complete.

• Data was protected and available only to the research team, primarily the principal investigator and statistician.

• Appropriate data analysis techniques were used to answer the research question.

Results

Line 78 What was the response rate? Does it mean it was 100%? Please state.

Response: Yes, the response rate was 100%. This is mentioned in the first line of the Results section.

Discussion

Please provide more details on strength and limitation, such as how limitation may not apply to the general public. In terms of limitation, since the study participants were only social media users, the findings cannot be generalized to the general population.

Please describe what proposed solution is possible based on the results of this study. It would be a very useful paper if specific solutions could be mentioned that would raise the level of knowledge of osteoporosis among the people of Pakistan.

Response: These are valid points and have been addressed in the manuscript. Thank you for your valuable review.

---

## [Decision Letter · Decision Letter 1]

12 May 2023

PONE-D-22-34888R1Assessing the Knowledge, Attitude and Practice of Osteoporosis among Pakistani Women: A National Social-Media Based SurveyPLOS ONE

Dear Dr. Ahmed,

Thank you for submitting your manuscript to PLOS ONE. After careful consideration, we feel that it has merit but does not fully meet PLOS ONE’s publication criteria as it currently stands. Therefore, we invite you to submit a revised version of the manuscript that addresses the points raised during the review process.

We look forward to receiving your revised manuscript.

Kind regards,

Tauqeer Hussain Mallhi, Ph.D

Academic Editor

PLOS ONE

Journal Requirements:

Additional Editor Comments:

Dear Authors, thank you for addressing the comments of the reviewers. The manuscript needs further improvement as referees are not much satisfied with the previous round of revisions.

Reviewers' comments:

Reviewer's Responses to Questions

**Comments to the Author**

1. If the authors have adequately addressed your comments raised in a previous round of review and you feel that this manuscript is now acceptable for publication, you may indicate that here to bypass the “Comments to the Author” section, enter your conflict of interest statement in the “Confidential to Editor” section, and submit your "Accept" recommendation.

Reviewer #1: (No Response)

Reviewer #2: All comments have been addressed

Reviewer #4: All comments have been addressed

2. Is the manuscript technically sound, and do the data support the conclusions?

Reviewer #1: (No Response)

Reviewer #2: Yes

Reviewer #4: Yes

3. Has the statistical analysis been performed appropriately and rigorously? 

Reviewer #1: (No Response)

Reviewer #2: Yes

Reviewer #4: Yes

4. Have the authors made all data underlying the findings in their manuscript fully available?

Reviewer #1: (No Response)

Reviewer #2: Yes

Reviewer #4: Yes

5. Is the manuscript presented in an intelligible fashion and written in standard English?

Reviewer #1: (No Response)

Reviewer #2: Yes

Reviewer #4: Yes

6. Review Comments to the Author

Reviewer #1: The paper has not been modified in line with suggestions. For example, discussion section.

Conclusion is lengthy. Kindly shorten it.

I think the word 'Demographic' has special meaning. It can be replaced with 'baseline information'.

Captions of tables need to be well described in detail. For example, for table number 1.

Reviewer #2: This study found that Adult Pakistani women have a fair understanding of osteoporosis, but the OPAAT tool clarifies some common misconceptions. There is a need to develop educational strategies to increase the knowledge of osteoporosis among Pakistani adults and to promote a bone-healthy lifestyle. This is an interesting and meaningful study, and I recommend accept.

Reviewer #4: Thank you for letting me review your article.

For the most part, It is much more coherent now that it has been revised according to the reviewer's advice.

Again, data management is very important in a social media study like this study. Please describe what you took care this time in the METHODOLOGY to manage the data so that future research using social media will not neglect data management in the same way.

7. PLOS authors have the option to publish the peer review history of their article (what does this mean?). If published, this will include your full peer review and any attached files.

Reviewer #1: **Yes: **Dr Satish Prasad Barnawal

Reviewer #2: No

Reviewer #4: No

While revising your submission, please upload your figure files to the Preflight Analysis and Conversion Engine (PACE) digital diagnostic tool, https://pacev2.apexcovantage.com/. PACE helps ensure that figures meet PLOS requirements. To use PACE, you must first register as a user. Registration is free. Then, login and navigate to the UPLOAD tab, where you will find detailed instructions on how to use the tool. If you encounter any issues or have any questions when using PACE, please email PLOS at figures@plos.org. Please note that Supporting Information files do not need this step.<quillbot-extension-portal></quillbot-extension-portal>

---

## [Author Response · Author response to Decision Letter 1]

6 Jun 2023

Authors’ Responses to Reviewer Comments 

Comments to the Author 

1. If the authors have adequately addressed your comments raised in a previous round of review and you feel that this manuscript is now acceptable for publication, you may indicate that here to bypass the “Comments to the Author” section, enter your conflict of interest statement in the “Confidential to Editor” section, and submit your "Accept" recommendation. 

Reviewer #1: (No Response) 

Reviewer #2: All comments have been addressed 

Reviewer #4: All comments have been addressed 

Authors’ Response: N/A 

2. Is the manuscript technically sound, and do the data support the conclusions? 

Reviewer #1: (No Response) 

Reviewer #2: Yes 

Reviewer #4: Yes 

Authors’ Response: N/A 

3. Has the statistical analysis been performed appropriately and rigorously? 

Reviewer #1: (No Response) 

Reviewer #2: Yes 

Reviewer #4: Yes 

Authors’ Response: N/A 

4. Have the authors made all data underlying the findings in their manuscript fully available? 

Reviewer #1: (No Response) 

Reviewer #2: Yes 

Reviewer #4: Yes 

Authors’ Response: N/A 

5. Is the manuscript presented in an intelligible fashion and written in standard English? 

Reviewer #1: (No Response) 

Reviewer #2: Yes 

Reviewer #4: Yes 

Authors’ Response: N/A 

6. Review Comments to the Author 

Reviewer #1: The paper has not been modified in line with suggestions. For example, discussion section. 

 Conclusion is lengthy. Kindly shorten it. 

 I think the word 'Demographic' has special meaning. It can be replaced with 'baseline information'. 

 Captions of tables need to be well described in detail. For example, for table number 1. 

Reviewer #2: This study found that Adult Pakistani women have a fair understanding of osteoporosis, but the OPAAT tool clarifies some common misconceptions. There is a need to develop educational strategies to increase the knowledge of osteoporosis among Pakistani adults and to promote a bone-healthy lifestyle. This is an interesting and meaningful study, and I recommend accept. 

Reviewer #4: Thank you for letting me review your article. 

For the most part, It is much more coherent now that it has been revised according to the reviewer's advice. 

 Again, data management is very important in a social media study like this study. Please describe what you took care this time in the METHODOLOGY to manage the data so that future research using social media will not neglect data management in the same way. 

Authors’ Response: 

To Reviewer 1: Thank you for your comments. The discussion has been modified in line with comments. Conclusion has been shortened and captions of tables have been described in detail. 

To Reviewer 2: Thank you for your comments! We appreciate the recommendation to accept. 

To Reviewer 4: Data management has been described in a separate section in the Methods.

---

## [Decision Letter · Decision Letter 2]

19 Jun 2023

Assessing the Knowledge, Attitude and Practice of Osteoporosis among Pakistani Women: A National Social-Media Based Survey

PONE-D-22-34888R2

Dear Dr. Ahmed,

We’re pleased to inform you that your manuscript has been judged scientifically suitable for publication and will be formally accepted for publication once it meets all outstanding technical requirements.

Kind regards,

Tauqeer Hussain Mallhi, Ph.D

Academic Editor

PLOS ONE

Additional Editor Comments (optional):

Reviewers' comments:

Reviewer's Responses to Questions

**Comments to the Author**

1. If the authors have adequately addressed your comments raised in a previous round of review and you feel that this manuscript is now acceptable for publication, you may indicate that here to bypass the “Comments to the Author” section, enter your conflict of interest statement in the “Confidential to Editor” section, and submit your "Accept" recommendation.

Reviewer #1: All comments have been addressed

Reviewer #4: All comments have been addressed

2. Is the manuscript technically sound, and do the data support the conclusions?

Reviewer #1: Yes

Reviewer #4: Yes

3. Has the statistical analysis been performed appropriately and rigorously? 

Reviewer #1: Yes

Reviewer #4: Yes

4. Have the authors made all data underlying the findings in their manuscript fully available?

Reviewer #1: Yes

Reviewer #4: Yes

5. Is the manuscript presented in an intelligible fashion and written in standard English?

Reviewer #1: Yes

Reviewer #4: Yes

6. Review Comments to the Author

Reviewer #1: I would like to congratulate the authors for wonderful work. It can be accepted for publication provided that all the technical and editorial procedures are over.

Reviewer #4: It has been corrected appropriately and is much better. This is an interesting and meaningful study, and I recommend accept.

7. PLOS authors have the option to publish the peer review history of their article (what does this mean?). If published, this will include your full peer review and any attached files.

Reviewer #1: **Yes: **Dr Satish Prasad Barnawal

Reviewer #4: No

<quillbot-extension-portal></quillbot-extension-portal>

---

## [Editor Report · Acceptance letter]

4 Jul 2023

PONE-D-22-34888R2 

Assessing the Knowledge, Attitude and Practice of Osteoporosis among Pakistani Women: A National Social-Media Based Survey 

Dear Dr. Ahmed:

I'm pleased to inform you that your manuscript has been deemed suitable for publication in PLOS ONE. Congratulations! Your manuscript is now with our production department. 

Kind regards, 

on behalf of

Dr. Tauqeer Hussain Mallhi 

Academic Editor

PLOS ONE